# Diagnosing Acute Cellular Rejection after Paediatric Liver Transplantation—Is There Room for Interleukin Profiles?

**DOI:** 10.3390/children10010128

**Published:** 2023-01-07

**Authors:** Imeke Goldschmidt, Evgeny Chichelnitskiy, Nicole Rübsamen, Veronika K. Jaeger, André Karch, Lorenzo D’Antiga, Angelo Di Giorgio, Emanuele Nicastro, Deirdre A. Kelly, Valerie McLin, Simona Korff, Dominique Debray, Muriel Girard, Loreto Hierro, Maja Klaudel-Dreszler, Malgorzata Markiewicz-Kijewska, Christine Falk, Ulrich Baumann

**Affiliations:** 1Department of Paediatric Liver, Kidney and Metabolic Diseases, Division of Paediatric Gastroenterology and Hepatology, Hannover Medical School, 30625 Hannover, Germany; 2Institute of Transplant Immunology, Hannover Medical School, 30625 Hannover, Germany; 3Institute of Epidemiology and Social Medicine, University of Münster, 48149 Münster, Germany; 4Paediatric Hepatology, Gastroenterology and Transplantation, Hospital Papa Giovanni XXIII, 24128 Bergamo, Italy; 5Liver Unit, Birmingham Children’s Hospital, Birmingham B4 6NH, UK; 6Department Pédiatrie, Services Spécialités Pédiatriques, Hôpitaux Universitaires de Genève, Rue Gabrielle-Perret-Gentil 4, 1211 Genève, Switzerland; 7Pediatric Liver Unit, Department of Paediatric Surgery, Hôpital Necker-Enfants malades, 75015 Paris, France; 8Servicio de Hepatologìa y Transplante, Hospital Infantil Universitario La Paz Madrid, 28046 Madrid, Spain; 9The Children’s Memorial Health Institute, 04-736 Warszawa, Poland

**Keywords:** paediatric, liver transplantation, immune monitoring, rejection, cytokine profile

## Abstract

Background: The current gold standard to diagnose T-cell-mediated acute rejection (TCMR) requires liver histology. Using data from the ChilSFree study on immune response after paediatric liver transplantation (pLT), we aimed to assess whether soluble cytokines can serve as an alternative diagnostic tool in children suspected to have TCMR. Methods: A total of n = 53 blood samples obtained on the day of or up to 3 days before liver biopsy performed for suspected TCMR at median 18 days (range 7–427) after pLT in n = 50 children (38% female, age at pLT 1.8 (0.5–17.5) years) were analysed for circulating cytokine levels using Luminex-based Multiplex technology. Diagnostic accuracy of cytokine concentrations was assessed using a multivariable model based on elastic net regression and gradient boosting machine analysis. Results: TCMR was present in 68% of biopsies. There was strong evidence that patients with TCMR had increased levels of soluble CXCL8, CXCL9, CXCL10, IL-16, IL-18, HGF, CCL4, MIF, SCGF-β, and HGF before biopsy. There was some evidence for increased levels of sCD25, ICAM-1, IL-6, IL-3, and CCL11. Diagnostic value of both single cytokine levels and a combination of cytokines and clinical markers was poor, with AUROCs not exceeding 0.7. Conclusion: Patients with TCMR showed raised levels of cytokines and chemokines reflective of T-cell activation and chemotaxis. Despite giving insight into the mechanisms of TCMR, the diagnostic value of soluble cytokines for the confirmation of TCMR in a clinical scenario of suspected TCMR is poor.

## 1. Introduction

Diagnosis of acute cellular rejection to date still requires liver biopsy. The decision that histological assessment is required is usually based on a combination of biochemical changes, such as persistently raised transaminases including gamma-glutamyl-transferase (GGT), as well as clinical signs and symptoms. Since these changes are non-specific, a biomarker tool that permits to guide the clinician’s decision regarding a liver biopsy or pre-emptive anti-rejection treatment would be invaluable.

To date, only the Cylex Immuknow assay has been evaluated as an immunological tool to diagnose acute cellular rejection [1]. The ImmuKnow measures T-cell reactivity in response to phytohemagglutinin-L. Its predictive value for acute cellular rejection is moderate at best [1]. It has indeed been found to be more suited to diagnose infection rather than rejection [1].

Data on the use of circulating cytokine levels to diagnose TCMR are scarce. Acute cellular rejection is widely attributed to an immunological imbalance with predominance of TH1 cytokines (TCMR) [2,3,4,5,6,7,8,9,10]. For the majority of publications, changes in circulating cytokines or intracellular cytokine expression before or early after transplantation were investigated as predictive for later occurrence of rejection [4,6,9,11,12], but not as diagnostic markers in the acute event of suspected rejection.

Studies that investigated cytokines directly before or during acute cellular rejection found circulating levels of IL-15 [13], IL-6 [14], IL-34 [15], IL-17 [12,16], and IL-23 [16] to be increased in patients with TCMR. Comparison in these cases was made with transplanted controls with either normal liver biochemistry, i.e., without the suspicion of acute rejection [12,13,14,15], or with patients without signs of rejection on an early protocol biopsy [16]. IL1-RA was found elevated 24 h preceding the diagnosis of TCMR not in plasma but in ascites in a study that monitored plasma and ascites cytokines up to 2 weeks after paediatric liver transplantation [17].

The international multicentric ChilSFree study investigates immune processes after paediatric liver transplantation in a prospective approach over the course of one year after pLT [18,19]. Its main aims are to improve pathophysiological understanding of the immune response after paediatric liver transplantation, as well as to identify markers suited to guide immunosuppressive therapy. With the current study, we aimed to address the clinical need to diagnose acute cellular rejection at a time of high inflammatory volatility. In order to address the potential of cytokine profiles as diagnostic tools for TCMR in a clinical situation where TCMR is suspected, we examined cytokine in a subset of patients from the ChilSFree study that underwent liver biopsy for suspicion of acute cellular rejection. By using patients with clinically suspected TCMR as non-rejection controls, we were able to address the potential diagnostic use of cytokine profiles in a real-life clinical scenario.

## 2. Materials and Methods

### 2.1. ChilSfree Study Design

In brief, patients undergoing de novo isolated liver transplantation received intensive immune monitoring over the course of one year. To this end, plasma samples were obtained before transplantation, at 1, 2, 3, and 4 weeks and 3, 6, and 12 months after transplantation. Additional samples were obtained at the time of liver biopsy. Clinical follow-up continued up to five years after transplantation.

### 2.2. Patients

Within the ChilSfree study, n = 246 liver biopsies were performed on clinical grounds. Of these, corresponding blood samples obtained up to 3 days before or on the day of the biopsy were available for n = 53 biopsies in n = 50 children. Demographical data and biopsy results are summarised in Table 1.

### 2.3. Immunosuppression

The immunosuppressive regimen followed local protocols in the participating centres. Details are given in Table 1. The majority of patients (66%) received steroid-free Tacrolimus-based immunosuppression, while 15% received Tacrolimus (TAC) with additional steroids per protocol. In n = 2 cases, steroids were added to TAC before the liver biopsy as the result of a previous biopsy-proven acute cellular rejection (BPAR). A combination of TAC and mycophenolate mofetil (MMF), or TAC, MMF, and steroids was used in 8% and 4% of cases, respectively. A total of 4% received cyclosporine (CSA) plus steroids. Mammalian target of rapamycin (mTOR) inhibitors were not used in this group.

### 2.4. Liver Biopsies

Biopsies were performed at a median of 18 days after transplantation (range 7–427 days). Indication for liver biopsy was made on clinical grounds by the local treating physician. Biopsies were read by the local attending pathologists. Degree of rejection was assessed by the Rejection Activity Index (RAI score).

### 2.5. Clinical Chemistry

Different participating centres used different analytical methods to determine clinical biochemistry parameters, with variations in units of measurements and the upper limit of normal. In order to ensure comparability of results, all biochemistry data are presented as ‘multiple of the upper limit of normal’ pertinent to the respective laboratory.

### 2.6. Soluble Immune Markers

Whole blood samples were collected at the participating centres and sent to a central laboratory (Institute for Transplant Immunology, Hannover Medical School, Hannover, Germany) for analysis. Transport to the central lab took an average of 2 days, during which time samples were stored at room temperature. After centrifugation, plasma samples for cytokine analysis were stored at −80 °C until further analysis. Soluble cytokine and chemokine levels were measured using a luminex-based multiplex approach (Bio-Rad, Hercules, CA, USA). At least 50 beads per analyte per sample or standard were recorded and the analysis was performed with the BioPlex Manager 6.1.1software.

### 2.7. Statistics

Continuous variables are presented as mean and standard deviation, or median plus range as appropriate. Skewed distribution was found for all cytokine results, and hence logarithmic transformation was performed before subsequent parametric analysis. Effect size of significant differences between the means was determined using Cohen’s d [20]. Frequencies of categorical variables were compared using either χ^2^ or Fisher’s exact test depending on patient numbers.

In order to assess the diagnostic accuracy of cytokine levels for the diagnosis of rejection, elastic net regression (ELR) and gradient boosting machine analysis were used independently to determine the best combinations of cytokine levels. For this, the sample cohort was divided into a development cohort (n = 34 samples taken between 2013 and 2015) and a validation dataset (n = 16 samples taken between 2016 and 2016). Area under the ROC curve (AUC) and calibration curves for the two models (ELR and GBM) were calculated in the validation dataset.

### 2.8. Ethical Considerations

This study was approved by the local Ethics Committee (Statement No. 1596-2012) and was in accordance with the Declaration of Helsinki on medical research involving human subjects. Informed consent was obtained from parents and caregivers.

## 3. Results

Acute cellular rejection was present in n = 36 (68%) of biopsies. RAI scores ranged from 2 to 9, with a median of 6. AST, ALT, GGT, and CRP values were markedly elevated in 53%, 69%, 83%, and 69% of patients, respectively, reflecting the hepatic dysfunction that prompted liver biopsy in the first place, as well as a high degree of inflammation. While the frequency of AST, ALT, GGT, Bilirubin, and CRP values above 1.5 times the upper limit of normal (ULN) was higher in patients with rejection, mean AST, ALT, GGT, and CRP values did not differ between the groups (Table 2 and Table 3). Mean bilirubin levels were higher in the rejection group (3.6 ± 3.9 × ULN (mean ± SEM) vs. 1.0 ± 1.2 × ULN, *p* < 0.01).

There was strong evidence that circulating levels of CXCL8, CXCL9, CXCL10, IL-16, IL-18, HGF, CCL4, MIF, SCGF-β, and HGF were elevated in the rejection group compared with the non-rejection group (Table 4). There was some evidence for increased levels of sCD25, ICAM-1, IL-6, IL-3, and CCL11, but these differences did not reach the levels of statistical significance.

The volcano plot analysis (Figure 1) shows that effects were largest for CXCL8 and CXCL9. Details on all measured cytokines are provided in Appendix A.

A total of n = 28 biopsies (53%) occurred within the first 3 weeks after pLT (Appendix A). Splitting the sample into early (<20 days post pLT) and late (≥20 days) yielded slightly different patterns of cytokine elevation compared with the whole cohort, with fewer cytokines marking the difference between rejectors and non-rejectors in late biopsies (Appendix A).

Circulating levels of IL-17, CXCL1, CXCL9, IL-12p70, CCL5, IL-12p40, FGF-b, IL-1, TNF-α, IL-9, TNF-β, VEGF, IL-4, and PDGF-β showed a positive correlation of moderate effect size with the degree of rejection as measured by RAI (Figure 2).

Diagnostic accuracy was first calculated for individual cytokines that had shown higher levels in rejection samples, with unsatisfactory results (data not shown).

An ELR with α = 0.1 and λ = 0.45 (determined via hyperparameter tuning) resulted in a model including sex, diagnosis, and 17 soluble cytokines (Table 5). CXCL8, CXCL9, IL-18, and IFN-α2 were also the most important features in the GBM model. In the validation dataset, the AUCs were low for both the ELR (AUC = 0.60) and the GBM (AUC = 0.64) model. Calibration was also poor in both models (Figure 3).

## 4. Discussion

We examined circulating cytokines and immune cell numbers in blood samples obtained at a maximum of three days prior to liver biopsy in n = 53 children after liver transplantation. All liver biopsies had been performed on clinical grounds, i.e., with the suspicion of acute cellular rejection based on laboratory and clinical features. Despite clear differences between rejection and non-rejection samples, neither single circulating cytokine levels nor a combination of cytokine levels showed sufficient predictive value to be considered as a reliable diagnostic test in a scenario of clinically suspected TCMR.

We found elevated levels of cytokines associated with T-cell activation (IL-16), interferon-γ-mediated inflammation (IL-18), immune cell chemotaxis and adhesion (CCL4, CXCL8, CXCL9, CXCL10, HGF), and regulators of innate immunity (MIF). In addition, IL-6, IL-3, CCL11, sCD25, and ICAM-1 were elevated in rejection samples compared to non-rejection samples, but these differences did not reach statistical significance.

IL-16 is a modulator of T-cell activation that also promotes cytokine secretion in monocytes and macrophages [21]. IL-16 deficiency reduced signs of rejection in a murine model for heart transplantation [22]. IL-18 induces IFNγ production. IL-18 suppression was associated with graft acceptance in rat models [23,24] and in a small human liver transplant case series [25]. CXCL9 and CXCL10 are IFNγ-induced endothelial chemokines involved in leucocyte adhesion during the early phases of post-transplant micro-inflammation [26]. Hepatocyte growth factor (HGF) regulates liver regeneration. Increased HGF mRNA expression in lymphocytes has been demonstrated after liver transplantation in patients with rejection as well as in patients with vascular causes for graft dysfunction [27], indicating that response to graft damage was not rejection specific. The observed patterns of elevated cytokines and chemokines in patients with rejection underscore the relevance of a pro-inflammatory micro-environment for the development of TCMR.

Our results differ in some points from previously published results [13,14,15,16]. In the existing literature, IL-15, IL-6, IL-34, IL-17, and IL-23 were found to be elevated during or shortly before acute cellular rejection [13,14,15,16]. Of these, IL-6, IL-15, IL-17, and the IL-23 subunit IL12p40 also were included in our panel but failed to discriminate between rejectors and non-rejectors.

IL-17 is produced by T-helper 17 (TH17) cells and has predominantly been implicated in the pathophysiology of autoimmune diseases [28]. IL-15 is produced by macrophages and increases proliferation of Th17 cells [29]. Elevated pre-transplant levels of IL-17 have been associated with early graft dysfunction, defined as elevated bilirubin, ALT, or INR [12], while elevation of IL-17 seven days post-transplant was predictive of early rejection [16]. Elevation of IL-15 has been associated with chronic and steroid-resistant rejection [13]. IL-6 has been repeatedly described as predictive of TCMR [5,12], but also is a strong marker of infection. IL-34 was not included in our panel.

Our study had a different aim compared to the quoted literature: we wanted to identify markers that can diagnose TCMR in a clinical scenario of suspected TCMR. Our non-rejection sample therefore consisted of patients with a strong clinical suspicion of TCMR, implying a certain degree of pre-existing inflammation. This is in sharp contrast to studies that used liver transplant recipients without signs of dysfunction, infection, or inflammation as controls [13,14,15,16]. It is well conceivable that the presence of non-TCMR-associated inflammation in our control group reduced the discriminatory value of cytokines involved in non-allogenic mediated inflammatory reactions [30]. However, the contrast in our study represents the situation of clinical decision making as it is and thus—different from previous studies—can evaluate the usefulness of biomarkers in clinical practice.

Neither single circulating cytokine levels nor a combination of cytokine levels showed sufficient predictive value to be considered as a valid diagnostic test in a scenario of clinically suspected TCMR. A number of features might explain this result. First, although this is one of the largest paediatric cohorts published that evaluates cytokines in post-transplant biopsies [6,11,17], numbers are still small. Evaluation of a diagnostic test requires evaluation in a training sample, with subsequent validation in a validation sample. The cohort therefore needs to be split into two groups for statistical analysis, thereby further reducing the available sample size and power. Second, the development of TCMR is a complex immunological process, which in the early days after transplantation is driven by a pro-inflammatory micro-environment that results from ischemia–reperfusion injury as much as from new alloantigen exposure [12,26]. About half of the biopsies in our study occurred within the first 2 weeks after transplantation, presumably fuelled by these mechanisms. A total of 21% of rejections occurred ‘late’, from 3 months up to a year after transplantation. It is conceivable that other mechanisms drive rejection at later time points, thereby obscuring the identification of a universal set of biomarkers. Indeed, when we split the sample according to early (<20 days) and late (≥20 days) biopsies, different patterns of elevated cytokines were observed with fewer pro-inflammatory changes in the later samples. However, these results have to be regarded with caution since splitting the sample further reduces the sample size. Third, potential bias might have been introduced by the fact the decision to perform liver biopsy in the first place, as well as the reading of the biopsies, was not centralised but was subject to the individual treating clinicians and pathologists at the participating centres.

While the low number of patients, albeit being one of the largest paediatric studies published for this topic, limits the interpretability of our statistical approach, the possibility to assess a whole panel of cytokines at the same time constitutes an obvious strength. Rather than focusing only on the well-established markers of Th1 cell activation such as IL-2 and IFNy, this approach allowed for the delineation of the important role played by pro-inflammatory chemokines in the promotion of TCMR.

## 5. Conclusions

Our study shows that circulating cytokine levels in peripheral blood have little diagnostic value in a clinical scenario of suspected rejection and, in patients with signs of hepatic dysfunction and inflammation, cannot safely discriminate rejectors from non-rejectors. Further studies are needed to decide whether early cytokine panel analysis is sufficiently precise to predict the development of later rejection and thus justify a pre-emptive increase in immunosuppressive therapy in selected patients.

## Figures and Tables

**Figure 1 children-10-00128-f001:**
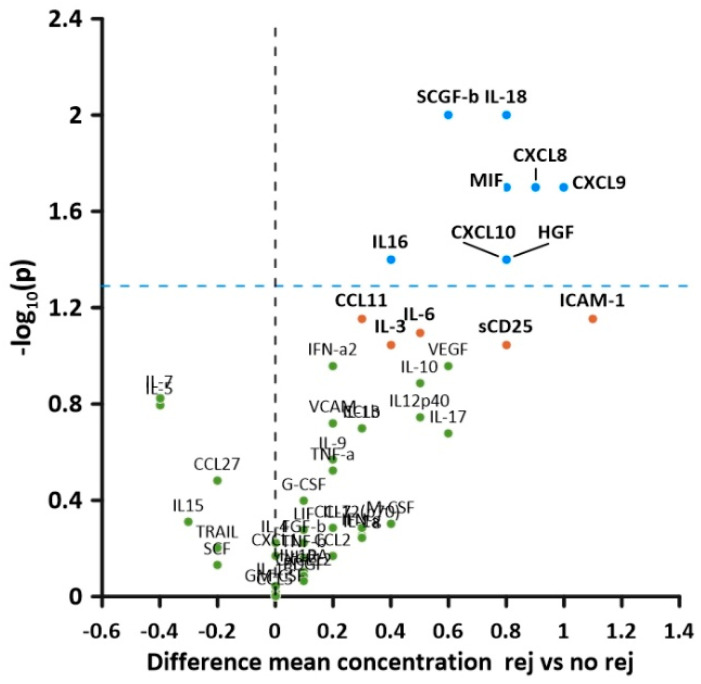
Volcano plot of soluble cytokine levels in patients with and without rejection. The *y*-axis represents −1× the logarithmised level of significance, while on the *x*-axis, the difference between mean concentrations in rejection (rej) and no rejection (no rej) samples is plotted. Cytokines above the blue plotted line had significantly higher values in patients with rejection vs. patients without rejection.

**Figure 2 children-10-00128-f002:**
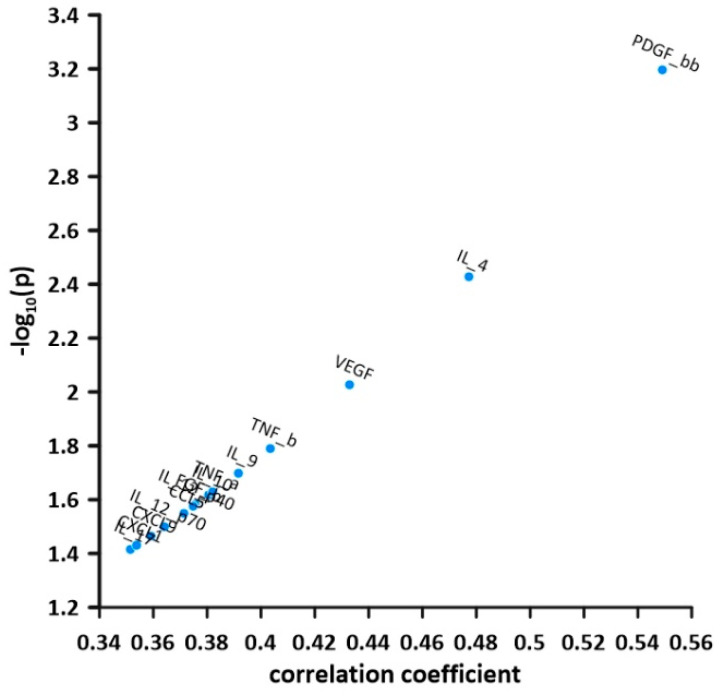
Volcano plot of the correlation of soluble cytokine levels with the degree of acute cellular rejection (correlation coefficient on the *x*-axis). The *y*-axis represents the −1× the logarithmised level of significance for each cytokine correlation. −log (*p* = 0.05) = 1.3.

**Figure 3 children-10-00128-f003:**
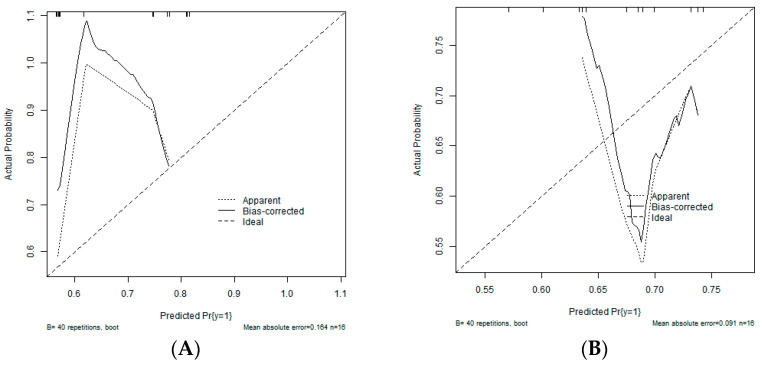
Calibration curves for the best fitting models. (**A**) Gradient boosting machine. (**B**) Elastic net regression.

**Table 1 children-10-00128-t001:** Demographic data of the study population.

		n (%)/Median (Range)
Sex	Boys	31 (62%)
	Girls	19 (38%)
Age at transplant		1.8 years (0.5–17.5)
Time lapse transplant to biopsy		18 days (7–427)
Primary disease	Biliary atresia	30 (60%)
	Acute liver failure	1 (2%)
	Metabolic liver disease	2 (4%)
	Sclerosing cholangitis	1 (2%)
	Autoimmune hepatitis	1 (2%)
	PFIC	3 (6%)
	Alagille syndrome	3 (6%)
	Hepatoblastoma	1 (2%)
	Other	8 (16%)
Living related liver transplantation		3 (6%)
ABO-incompatible transplantation		3 (6%)
Liver biopsies	Acute cellular rejection	36 (67.9%)
	No rejection	17 (32.1%)
RAI scores		Median 6 (2–9)
	2	2 (5.7%)
	4	3 (8.6%)
	5	7 (20.0%)
	6	13 (37.1%)
	7	4 (11.4%)
	8	5 (14.3%)
	9	1 (2.9%)
Immunosuppression at time of biopsy		
	Tacrolimus monotherapy	35/ 53 (66.0%)
	Tacrolimus + steroids *	10/53 (18.9%)
	Tacrolimus + MMF	4/53(7.5%)
	Tacrolimus + MMF + steroids	2/53(3.8%)
	Cyclosporine + steroids	2/53 (3.8%)

* Steroids per local protocol n = 8, steroids following preceding rejection n = 2. MMF: mycophenolate mofetil.

**Table 2 children-10-00128-t002:** Biochemistry at time of biopsy.

	All ^1^	No Rejection	Rejection	*p*
	Mean ± SD	Mean ± SD	Mean ± SD	
AST (times ULN)	2.1 ± 1.9	1.8 ± 2.3	2.2 ± 1.7	0.43
ALT (times ULN)	3.4 ± 2.8	2.5 ± 2.4	3.8 ± 2.9	0.11
GGT (times ULN)	7.7 ± 12.3	5.2 ±5.5	8.8 ± 14.3	0.34
Bilirubin (times ULN)	2.8 ± 3.5	1.0 ± 1.2	3.6 ± 3.9	<0.01
CRP (times ULN)	6.6 ± 8.7	3.7 ± 5.6	7.5 ± 9.3	0.23

^1^ Values available in n = 52 (GGT, ALT, Bilirubin), n = 51 (AST), and n = 41 (CRP).

**Table 3 children-10-00128-t003:** Frequency of values > 1.5 times upper limit of normal.

	All	No Rejection	Rejection	*p* ^1^
AST	27/51 (52.9%)	5/16 (31.3%)	22/35 (62.9%)	0.04
ALT	38/52 (73.1%)	8/16 (50.0%)	30/36 (83.3%)	0.01
GGT	43/52 (82.7%)	10/16 (62.5%)	33/36 (91.7%)	0.03
Bilirubin	20/52 (38.5%)	3/16 (18.8%)	17/36 (47.2%)	0.048
CRP	29/42 (69.0%)	5/11 (45.5%)	24/31 (77.4%)	0.049

^1^ χ^2^/Fisher’s exact test.

**Table 4 children-10-00128-t004:** Cytokine levels in rejection and non-rejection samples.

	Rejection Ln (conc) (pg/mL) mean ± SD	No Rejection Ln (conc) (pg/mL) mean ± SD	T (df)	*p*	Effect Size Cohen’s d
CXCL8	3.1 ± 1.4	2.2 ± 0.8	2.4 (50)	0.02	1.2
CXCL9	7.0 ± 1.0	6.0 ± 1.8	2.4 (22)	0.02	1.3
CXCL10	6.4 ± 1.2	5.6 ± 1.1	2.2 (50)	0.04	1.2
IL-16	6.3 ± 0.8	5.9 ± 0.7	2.1 (50)	0.04	1.1
IL-18	5.0 ± 1.1	4.2 ± 1.2	2.6 (51)	0.01	1.1
CCL4	4.1 ± 0.6	3.8 ± 0.4	2.2 (50)	0.03	0.5
MIF	8.2 ± 1.2	7.4 ± 1.0	2.4 (51)	0.02	1.1
SCGF-b	10.0 ± 0.7	9.4 ± 1.0	2.5 (51)	0.01	0.8
HGF	6.4 ± 1.3	5.6 ± 1.1	2.2 (51)	0.04	0.52
sCD25	6.3 ± 1.4	5.5 ± 1.5	1.7 (51)	0.09	1.4
ICAM-1	10.9 ± 0.5	9.8 ± 2.3	1.9 (16.8)	0.07	1.4
IL-6	2.5 ± 1.1	2.0 ± 1.0	1.8 (50)	0.08	1.1
CCL11	3.7 ± 0.7	3.4 ± 0.6	1.8 (50)	0.07	0.7
IL-3	5.9 ± 0.7	5.5 ± 0.8	1.7 (51)	0.09	0.8

**Table 5 children-10-00128-t005:** Cytokines included into a diagnostic model derived from elastic net regression.

Variable	Coefficient	Variable	Coefficient
(Intercept)	−0.9172	CCL11_log	0.0511
Sex: female	0.0774	CCL27_log	−0.0059
Diagnosis: acute liver failure	−0.2218	CXCL1_log	0.0214
IL_1b_log	0.0094	CXCL8_log	0.0356
IL_3_log	0.0135	CXCL9_log	0.0254
IL_7_log	−0.0215	SCGF_b_log	0.0113
IL_16_log	0.0326	MIF_log	0.0219
IL_18_log	0.0318	sCD25_log	0.0007
IFN_a2_log	0.0405	ICAM_1_log	0.011
LIF_log	0.002	CCL4_log	0.0223

## Data Availability

Data will be made available on request by the corresponding author.

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
