# Peer review of "Diagnosing Acute Cellular Rejection after Paediatric Liver Transplantation—Is There Room for Interleukin Profiles?"

_children, 2023, doi:10.3390/children10010128_

Round 1

Reviewer 1 Report

      The article by Goldschmidt et al may be advisable for publication but after proofreading. A limitation may be the low number of study samples. However, to improve the article I suggest the following corrections: There are lexical errors, for example, Tracolimus has been replaced by tTCRolimus in tables and text. Other lexical errors such as inflammation mTCMophages (it will be macrophages). Finally, it would have been interesting to study intracellular rather than serological cytokines. In subsequent studies the group should study them and compare them with the serological ones. Finally you should name an article that is missing from your bibliography. You should name it:

Cytokine Expression Profile as Predictive Surrogate Biomarkers for Clinical Events in the Field of Solid Organ Transplantation. Boix F, Mrowiec A, Muro M. Curr Protein Pept Sci. 2017;18(3):240-249. doi: 10.2174/1389203717666160902130001.

Author Response

Thank you for your kind review. Thank you for pointing out the lexical errors, which we corrected. We agree that it would indeed be fascinating to compare peripheral cytokines with intracellular cytokine staining from liver tissue, as well as with immune cell staining. This is planned for a future analysis. Thank you for pointing out the review by Boix et al, which we referenced in the introduction and added to the bibliography.

Reviewer 2 Report

The study of Goldschmidt et al. evaluated whether circulating cytokine levels can accurately predict/identify TCMR in a cohort of transplanted pediatric patients from the CHilSFree clinical trial. They quantified the concentration of several rejection-related cytokines with the Luminex-based multiplex assay and assessed the corresponding diagnostic potential with a multivariable model. 

This is an interesting study addressing a relevant topic with a potential clinical impact; the Authors clearly described their findings and the study limitations and discussed their data with proper research and clinical contextualization.  

In further studies, it will be interesting to more granularly stratify the RAI score, as it takes into account different types of allograft injury (e.g., test whether cytokines levels are related to portal inflammation, bile duct injury, or endotheliitis).

Author Response

Thank you for your kind review. We agree that it would be interesting to stratify according to RAI subscores, as well as look for intrahepatic cytokine and immune cell staining to get a more thorough understanding. Within the current data set, there is a limit in numbers potentially precluding a meaningful analysis from more detailed subgroup division. We will therefore happily take up this proposition for future analyses.

Reviewer 3 Report

This is important topic. Many of us have tried to find useful biomarkers for rejection and most have failed. It does't mean that one shouldn't try.

Authors have a good study design. I would suggest that they should even more clearly indicate the study design at intro emphasizing the fact that they are testing if cytokine profile is helpful in situation when rejection is suspected, rather than it is useful in rejection diagnosis in general. In addition, intro or methods section would improve if ChilSFree study design is briefly opened in this paper also.

This study has some limitations which are pointed out also by the authors. Wide time lapse transplant to biopsy (mean 18 days (7-427!!)) is one of those limitations pointed out. However, to my mind this data should be showed  in more detail ( early vs late separately?). 

Some explanation why mean  and SEM are presented not mean and SD?

Why labs are compared as Frequency of values > 1.5 times upper limit of normal? Why not as continuous variables?

Minor typos etc:

tTCMRolimus (TAC) line 90 and many times later

inflammataion.. line 141

mTCMRophages line 202
